# Genome-Wide Identification and Comprehensive Analysis of the *FtsH* Gene Family in Soybean (*Glycine max*)

**DOI:** 10.3390/ijms242316996

**Published:** 2023-11-30

**Authors:** Qi Shan, Baihui Zhou, Yuanxin Wang, Feiyu Hao, Lin Zhu, Yuhan Liu, Nan Wang, Fawei Wang, Xiaowei Li, Yuanyuan Dong, Keheng Xu, Yonggang Zhou, Haiyan Li, Weican Liu, Hongtao Gao

**Affiliations:** 1Engineering Research Center of the Chinese Ministry of Education for Bioreactor and Pharmaceutical Development, College of Life Sciences, Jilin Agricultural University, Changchun 130118, China; sq1806006978@163.com (Q.S.); 15754366701@163.com (B.Z.); 17833122454@163.com (Y.W.); hao2769631240@163.com (F.H.); 17644016346@163.com (L.Z.); yuhanlnn@163.com (Y.L.); nanw@jlau.edu.cn (N.W.); wangfawei@jlau.edu.cn (F.W.); xiaoweili1206@jlau.edu.cn (X.L.); yuanyuand@jlau.edu.cn (Y.D.); 2Sanya Institute of Breeding and Multiplication, School of Breeding and Multiplication, Hainan University, Sanya 572025, China; xh312319@163.com (K.X.); ygzhou@hainanu.edu.cn (Y.Z.); hyli@hainanu.edu.cn (H.L.)

**Keywords:** soybean, *FtsH* family, chloroplast, protein hydrolysis enzyme

## Abstract

The filamentation temperature-sensitive H (*FtsH*) gene family is critical in regulating plant chloroplast development and photosynthesis. It plays a vital role in plant growth, development, and stress response. Although *FtsH* genes have been identified in a wide range of plants, there is no detailed study of the *FtsH* gene family in soybean (*Glycine max*). Here, we identified 34 *GmFtsH* genes, which could be categorized into eight groups, and *GmFtsH* genes in the same group had similar structures and conserved protein motifs. We also performed intraspecific and interspecific collinearity analysis and found that the *GmFtsH* family has large-scale gene duplication and is more closely related to *Arabidopsis thaliana*. Cis-acting elements analysis in the promoter region of the *GmFtsH* genes revealed that most genes contain developmental and stress response elements. Expression patterns based on transcriptome data and real-time reverse transcription quantitative PCR (qRT-PCR) showed that most of the *GmFtsH* genes were expressed at the highest levels in leaves. Then, GO enrichment analysis indicated that *GmFtsH* genes might function as a protein hydrolase. In addition, the GmFtsH13 protein was confirmed to be localized in chloroplasts by a transient expression experiment in tobacco. Taken together, the results of this study lay the foundation for the functional determination of *GmFtsH* genes and help researchers further understand the regulatory network in soybean leaf development.

## 1. Introduction

As the primary site of photosynthesis in plants, maintaining the internal homeostasis of chloroplasts is crucial for plant growth and development. However, due to their specific semi-autonomous nature, the proteins within them are encoded by both nuclear and chloroplast genes [1], which leads to the need for the reprocessing of most proteins as well as the degradation of excess and damaged proteins [2]. As key players in balancing protein synthesis and hydrolysis, chloroplast proteases are essential for photosynthesis and other biochemical processes within the chloroplast [3]. In higher plants, three major chloroplast proteases have been identified: Clp (caseinolytic protease), Deg (periplasmic protein degradation), and FtsH (filamentation temperature-sensitive H) [4,5]. Among them, FtsH protease is an ATP-dependent metalloprotease bound to the endosome with proteolytic activity and molecular chaperone activity [6]. The *FtsH* gene was initially discovered and named after the temperature-sensitive fibrillar mutant *ftsh1* obtained through chemical mutagenesis [7]. The *FtsH* gene belongs to the AAA+ protease family and has three conserved structural domains: the N-terminal transmembrane structural domain, the C-terminal ATPase structural domain, and the M41 protein hydrolase structural domain [8]. The metalloprotease structural domain, which contains a zinc-binding sequence (HEXXH), is the active center of protease hydrolysis and is required for its catalytic activity [9]. The FtsH protein works by oligomerizing to form a hexameric complex [10], which encapsulates the active site of protein hydrolysis in the polymer’s center and renders individual subunits inactive [11].

The plant *FtsH* gene was first identified in spinach leaves [12] and subsequently found in several species, including tobacco [13], Arabidopsis [14], and rice [15]. Its encoded proteins are usually localized in chloroplasts and mitochondrial membranes. Most members have been shown to affect chloroplast development and homeostatic homeostasis, including involvement in biological processes such as vesicle formation [16], PS II photodamage repair [17], early assembly of the PS I complex [18], the degradation of cytochrome b6f complex subunits [19], and ATP synthase assembly under heat stress [20] and have a direct effect on plant phenotype changes. Previous research discovered that deleting *AtFtsH2* (*var2*) and *AtFtsH5* (*var1*) resulted in an increase in reactive oxygen species. The formation of albino spotted leaves in Arabidopsis [21], while knockdown of *AtFtsH12* and *AtFtsHi1* directly led to Arabidopsis embryonic death [22,23]; not only that, but similar phenotypes were found in other species: the *NtFtsH2*- and RNA-interfering lines of *NtFtsH1* exhibit yellowish, heterochromatic tobacco leaves; *osftsh2* knockout mutant rice not only produces albino leaves but also fails to grow to the trifoliate stage [15]. Furthermore, FtsH protease is important for increasing crop yield [24], heat tolerance [20], salt tolerance, and drought tolerance [25].

Soybean is an important oilseed cash crop that belongs to the legume family of annual plants and is grown primarily in China, India, the United States, Brazil, and Argentina [26]. Several factors affect its growth, developmental status, and yield, the most important of which are photosynthesis and light levels [27]. Previous research has shown that increasing the expression of photosynthesis-related genes improves photosynthesis in soybeans while increasing seed yield [28,29]. However, the specific regulatory mechanisms controlling photosynthesis in soybean leaves remain largely unknown, especially the *FtsH* family-mediated pathway. As a result, the *FtsH* gene family of *Glycine max* was thoroughly examined in this study, including analyses of phylogenetic relationship, gene structure, motif composition, promoter elements, chromosome localization, protein structure, and GO enrichment. The evolutionary relationships between soybean and other species, as well as the expression profiles in different tissues and the subcellular localization, were then investigated further. This study provides insight into the function of the *GmFtsH* gene family and hopes to provide a basis for future mechanistic studies of the *GmFtsH* genes in soybean leaf development.

## 2. Results

### 2.1. Identification of the FtsH Gene Family in Soybean (Glycine max)

To identify the *FtsH* gene family in *Glycine max*, we searched the Phytozome database (*Glycine max* Wm82.a2.v1) using *AtFtsH* gene sequences and obtained 34 candidate sequences. Additionally, we used CD-search and Pfam to look for genes that had the structural domains AAA (PF00004) and peptidase M41 (PF01434), which are exclusive to the *FtsH* gene family. Finally, we obtained 34 *FtsH* gene family members of *Glycine max*, reported as *GmFtsH1-25* and *GmFtsHi1-GmFtsHi9* (without a Zn^2+^-binding module) [24]. According to sequence analysis, the 34 GmFtsHs have proteins with lengths ranging from 228 (GmFtsH12) to 1288 amino acids (GmFtsHi4); their molecular weights range from 25,504.24 (GmFtsH12) to 147,844.1 kD (147,844.1); their isoelectric points range from 5.01 (GmFtsH23) to 9.55 (GmFtsHi8), of which the acidic proteins accounted for 75%, indicating that most GmFtsHs are rich in caustic bases; the instability coefficients were 29.58–50.51. The most stable protein was GmFtsH5, which had a stability index of 29.58. All members’ hydrophobicity indices were negative (−0.545–−0.077), indicating varied degrees of hydrophilicity in all GmFtsH proteins, and the lipolysis index ranged from 81.4 to 96.11, indicating that all GmFtsH proteins were lipolysis proteins. The predicted results of subcellular localization showed that most GmFtsHs were located in mitochondria and extracellular tissues, except for GmFtsH12, which was expressed in chloroplasts (Table 1).

### 2.2. Phylogenetic Analysis of GmFtsH Genes

To investigate the phylogenetic relationships of GmFtsH genes, we constructed phylogenetic trees using FtsH protein sequences from *Arabidopsis thaliana* (17), *Oryza sativa* (9), *Nicotiana tabacum* (20), and *Glycine max*. By comparing the full-length protein sequences of 80 *FtsH* genes from the four species, these genes were divided into eight groups, which is consistent with the classification of *FtsH* in *Pyrus bretschneideri* [30]. We named these Groups I–VIII, and each group’s genes had their own specific highly conserved amino acid sequences (Figure 1 and Appendix A). Each group contained 22, 13, 11, 14, 8, 4, 4, and 4 *FtsH* genes, respectively. *FtsH* genes from the four species were present in Groups I, II, III, and IV. This result, along with multiple sequence alignment results, suggests that these members of the four plant species may have undergone similar evolutionary patterns. There was also an unequal distribution in the evolutionary tree, with more *AtFtsHs* and *GmFtsHs* clustered together and similarities in the gene structure of members of the same subfamily, suggesting that the evolutionary relationship of Arabidopsis and soybean *FtsH* genes may be closer. *AtFtsHs* were distributed in Groups I to VIII, while most *OsFtsHs* were concentrated in Groups I and IV and were more distantly related to most of the genes, suggesting that *GmFtsH* genes may have experienced similar evolutionary patterns. *AtFtsHs* were distributed from Group I to Group VIII. In contrast, *OsFtsHs* were primarily concentrated in Group I and Group IV and evolved more distantly from most genes, suggesting that *GmFtsHs* were more closely related to *AtFtsHs* and *NtFtsHs* than to *OsFtsHs*, which may be because *Glycine max*, *Arabidopsis thaliana*, and *Nicotiana tabacum* are dicotyledonous plants.

### 2.3. Gene Structure and Conserved Motif Analysis of the GmFtsH Gene Family

To further determine the structural differences and evolutionary relationships among *GmFtsH* genes, we constructed a phylogenetic tree and analyzed the gene structures, conserved structural domains, and motifs of the *GmFtsH* genes (Figure 2). Gene structure analysis showed that all *GmFtsHs* contained multiple introns and exons (Figure 2A,B), and the number of exons ranged from 3 (*GmFtsH23*) to −27 (*GmFtsH15*). Among them, 31 *GmFtsH* genes had an untranslated region (UTR), and *GmFtsH* genes in the same group had similar gene structures, suggesting that *GmFtsH* genes have a highly conserved exon-intron structure. We analyzed the conserved structural domains of *GmFtsH* genes by TBtools and, according to the results, all *GmFtsH* genes contained FtsH-related specific structural domains (ftsH, FtsH_fam superfamily, FtsH_ext) (Figure 2D), and other conserved domains existed in individual genes, which might be related to their gene functions. The ten conserved motifs (Motif1-10) of the *GmFtsH* genes were further predicted using the MEME website (Figure 2C), and the results showed that 21 *GmFtsH* genes contained all the conserved motifs, and all members except *GmFtsH16* contained Motif3, which suggests that this motif can be used as a marker for the identification of *FtsH* genes. Cooperative phylogenetic tree analysis revealed that the number and distribution of motifs were similar among members of the same subfamily, and the distribution of motifs among the sequences of genes distributed in different subfamilies varied considerably, and there might be differences in gene function as well. 

### 2.4. Chromosome Location and Collinearity Analysis

Based on the soybean genome annotation, we mapped the physical localization of *GmFtsH* genes on soybean chromosomes. As shown in Appendix A, 34 *GmFtsH* genes were unevenly distributed on 17 chromosomes, among which *GmFtsH11* was distributed on Scaffold_25, which might be caused by *FtsH* gene duplication during soybean evolution. Gene duplication is a major factor in increasing the number of genes and evolutionary innovations, so we analyzed gene duplication events in *GmFtsH* genes. *GmFtsH6* and *GmFtsH7* located on chromosome 12 (Gm12) and *GmFtsH19*, *GmFtsH20*, and *GmFtsH21* on chromosome 14 (Gm14) were tandemly duplicated sequences. We further analyzed the duplication events of the *GmFtsH* genes by covariance analysis using MCScanXs. The results showed 19 covariant gene pairs generated by genome-wide duplication events in the *GmFtsH* gene family. The paired covariant genes may have similar biological functions (Figure 3). To trace the origin and evolution of the *GmFtsH* gene family, we constructed a covariance map of *Glycine max* with *Arabidopsis thaliana*, *Oryza sativa*, *Nicotiana attenuata*, and *Solanum lycopersicum* (Figure 4). The results showed that 21 pairs of covariance relationships were found in the covariance analysis of soybean genes and Arabidopsis genes, eight pairs of covariance relationships were found with rice, 20 pairs of covariance relationships were found with tomato, and only 2 pairs of covariance relationships were found with tobacco. We thus speculate that the *FtsH* gene family of soybean is more closely related to Arabidopsis, which is consistent with the results of the phylogenetic tree analysis.

### 2.5. Cis-Acting Elements Analysis of GmFtsH Genes

The 2000 bp upstream promoter sequence of the *GmFtsH* genes’ start site was utilized, and the PlantCARE online database was used to analyze the cis-acting elements of the *GmFtsH* genes. The prediction results showed that all *GmFtsHs* contained many CAAT boxes and TATA boxes, proving that they could be transcribed usually. Furthermore, they contained five classes of stress-related elements, including anaerobic-inducibility elements (71), low-temperature responsiveness elements (20), defence and stress responsiveness elements (17), drought-inducibility elements (14), and anoxic-specific inducibility elements (5); eight classes of developmental associated elements, including light-response elements (423), cell cycle regulation elements (3), seed-specific regulation elements (3), meristem expression elements (25), endosperm expression elements (6), circadian control elements (7), zein metabolism regulation elements (18), and palisade cells differentiation elements (1); and five classes of hormone-related elements, including abscisic acid responsiveness elements (76), MeJA responsiveness elements (66), gibberellin responsiveness elements (28), Auxin responsiveness elements (17), and salicylic acid responsiveness elements (20). Among these cis-elements, light-response elements (135) accounted for the largest category, followed by abscisic acid responsiveness elements (76), which accounted for the second largest category. Interestingly, the promoters of *GmFtsH* genes contain many elements involved in MeJA responsiveness elements, suggesting that the function of *GmFtsH* genes may be related to the damage response. In addition, the vast majority of *GmFtsH* genes contained MYB binding sites (239) and MYC binding sites (139) (Figure 5, Appendix A). The above results suggest that *GmFtsH* genes may affect soybean growth and development, various hormones, and stress-related responses.

### 2.6. Gene Ontology (GO) Enrichment Analysis

Gene ontology (GO) enrichment analysis is a standard method for gene function analysis, so we performed GO enrichment analysis of *GmFtsH* gene family members. The results showed that the *GmFtsH* genes might be involved in various biological, cellular, and molecular processes, including five biological process terms, seven cellular component terms, and six molecular function terms (Figure 6, Appendix A). The three most abundant molecular process terms were “ATP-dependent peptidase activity”, “metallopeptidase activity”, and “ATPase activity”, suggesting that the *GmFtsH* genes may have different enzymatic activities to regulate molecular processes. It is worth noting that the most enriched category of cellular components was “chloroplast vesicle” and the most enriched category of biological processes was “protein hydrolysis”. These results suggest that *GmFtsH* genes may function as enzymes in chloroplast-like vesicles for protein hydrolysis biology.

### 2.7. Expression Pattern of FtsH Genes in Glycine max under Different Tissues

To investigate the role of *GmFtsH* genes in the growth and development of soybeans, we analyzed the expression pattern of each member of the *GmFtsH* gene family based on gene expression matrices downloaded from the Phytozome genome database for a variety of organs, including leaves, nodules, root hairs, pods, stems, seeds, shoot apical meristems, flowers, and roots (Figure 7, Appendix A). The results showed that *GmFtsH* genes were expressed differently in different tissues of soybean. We found that the expression patterns of *GmFtsHi1-GmFtsHi9* and *GmFtsH22*; *GmFtsH24* and *GmFtsH25*; *GmFtsH20* and *GmFtsH21*; *GmFtsH13*, *GmFtsH14*, and *GmFtsH15*; and *GmFtsH1*, *GmFtsH2*, *GmFtsH3*, and *GmFtsH4* were more similar. *GmFtsHi1-GmFtsHi9* and *GmFtsH22* were expressed at higher levels in all tissues except roots. *GmFtsH24* and *GmFtsH25* were both expressed at lower levels in seeds and shoot apical meristems. *GmFtsH20* and *GmFtsH21* were expressed at lower levels in stems. *GmFtsH13*, *GmFtsH14*, and *GmFtsH15* all had low expression in roots, nodules, and root hairs and higher expression in leaves and flowers. *GmFtsH1*, *GmFtsH2*, *GmFtsH3*, and *GmFtsH4* were highly expressed in leaves, had low expression in roots and nodules, and were moderately expressed in other tissues. In addition, *GmFtsH23* was not expressed in all tissues, *GmFtsH17* was not expressed in all tissues except seeds, and all members except *GmFtsH5*, *GmFtsH6*, *GmFtsH12*, *GmFtsH23*, and *GmFtsH17* were significantly highly expressed in leaves. Most of the *GmFtsH* genes showed low expression in roots. From this, we hypothesized that different *GmFtsH* genes may play different functions in affecting soybean development, but most *GmFtsH* genes may play essential roles in soybean leaf development.

In addition, based on the results of phylogenetic relationship analyses, we selected 14 of the 34 *GmFtsH* genes to further determine the expression patterns of *GmFtsH* genes. *GmFtsH1*, *GmFtsH2*, *GmFtsH5*, *GmFtsH10*, *GmFtsH11*, *GmFtsH12*, *GmFtsH13*, *GmFtsH16*, *GmFtsH18*, *GmFtsH24*, *GmFtsHi1*, *GmFtsHi4*, *GmFtsHi5*, and *GmFtsHi6* were expressed in nine tissues (new leaf, old leaf, shoot apical meristem, stem, main root, lateral root, flower, seed pod, and seed embryo) with relative expression levels. The results showed that the qRT-PCR expression patterns of most *GmFtsH* genes were consistent with the general trends of transcriptome expression patterns, with almost identical results for *GmFtsH11*, *GmFtsH18*, *GmFtsH24*, and *GmFtsHi6* (Figure 8). We found that all genes except *GmFtsH13* were highly expressed in new and old leaves, and *GmFtsH1*, *GmFtsH5*, *GmFtsH10*, *GmFtsH11*, *GmFtsH13*, *GmFtsH18*, *GmFtsH24*, and *GmFtsHi5* were moderately expressed in flowers. However, most genes were expressed at low levels in the shoot apical meristem tissue, which differed from the transcriptome data, which may have been due to sample differences in sequencing and qRT-PCR.

### 2.8. Subcellular Localization Analysis of GmFtsH13

According to the predicted subcellular localization of GmFtsH proteins, these proteins were located in the mitochondria, chloroplast, and extracellular tissues. The *GmFtsH13* gene has been found to share biological roles with the *AtFtsH1* and *AtFtsH5* genes through examination of the gene’s structure, conserved domains, and phylogenetic tree. Therefore, we amplified the *GmFtsH13* gene and built it into the plant expression vector pGDG containing a green fluorescent protein tag (mGFP) driven by the 35S promoter for the transient transformation of tobacco to clarify its subcellular localization in order to further validate the results of the subcellular localization prediction. The findings demonstrated that the GmFtsH13 protein was localized in chloroplasts since the GmFtsH13-mGFP green fluorescence signal coincided with the chloroplast autofluorescence signal and was concentrated there (Figure 9).

## 3. Discussion

As chloroplast proteases, FtsH proteins are crucial for plant growth, development, and resistance to environmental stress, particularly when it comes to controlling chloroplast formation [14,20,31,32]. Recent research on the soybean plant has demonstrated that *GmFtsH25* increases seed output through improved photosynthesis, and it was discovered that the *GmFtsH9* gene may have a role in controlling PS II activity [33]. As a result, it is essential to thoroughly examine the roles and molecular processes by which the *GmFtsH* gene family affects soybean development and stress resistance. To date, the *FtsH* gene family has been identified in various types of plants, such as tobacco [13], Arabidopsis, rice [34], maize [35], and pear [30]. However, additional knowledge is required regarding the *FtsH* gene family in *Glycine max*.

To better understand the biological functions of the *GmFtsH* gene family, we performed bioinformatics analysis and expression pattern analysis. In this study, we identified 34 *FtsH* family members of *Glycine max*, named *GmFtsH1-25* and *GmFtsHi1-i9*, all of which contained FtsH-associated conserved structural domains. Phylogenetic analysis was also performed on the soybean, Arabidopsis, rice, and tobacco *FtsH* gene families and these members were divided into eight groups, which is consistent with the classification of *FtsH* genes in Arabidopsis and pear [30,34]. The findings from the analyses of gene structure and conserved motifs further supported the higher degree of sequence similarity among *GmFtsH* genes from the same groups.

Gene duplication contributes additional genes and genetic material to the body of the plant, which is a necessary mechanism for species development and evolution [36]. In the *GmFtsH* gene family, the current investigation discovered 19 sets of fragment duplication events and 2 sets of tandem duplication events. Compared to previous studies in tobacco and pear [13,30], the *GmFtsH* family gene duplication was much more significant, especially fragment duplication, and had more members, almost four times the number of members in rice (9). This extensive replication may be connected to two polyploid soybean genome duplications that occurred about 59 and 13 million years ago [37]. The two duplication events were followed by genetic diversification and extensive chromosomal rearrangements, resulting in highly replicated genomes. This shows that the *FtsH* gene family’s proliferation is dependent on instances of gene duplication and connected to the species itself.

With the continuous development of and improvement in biological sciences, identifying specific genes and their functions is crucial for the survival of organisms, and gene ontology is a vital method to structure the knowledge of gene functionality [38], so we performed GO enrichment analyses of GmFtsH proteins. According to the gene enrichment results, most GmFtsH proteins are found in chloroplasts and mitochondria, which are important organelles, as well as in cell membranes. They participate in biological processes including digestion and protein hydrolysis. Previous studies have also confirmed the involvement of FtsH proteins in the degradation of proteins related to the photosynthetic system. For instance, suppressing *NtFtsH1* and *NtFtsH2* expression decreased PSII activity and postponed the degradation of the D1 protein, disturbing the cystoid membrane and resulting in the formation of heterochromatic tobacco leaves [39]. In Chlamydomonas sulphur or nitrogen deficiency, FtsH degraded the cytochrome b6f complex, controlling protein quality during nutrient deficiency [19]. And the results of the GO enrichment analysis suggest that the *FtsH* gene is involved in embryonic development at the end of seed dormancy, which may be responsible for the lethal phenotype of Arabidopsis embryos following FtsH deletion [23,40]. Furthermore, the molecular functional analysis showed that GmFtsH proteins have ATP-dependent peptidase activity and chloroplast protein-transporting ATPase activity, which is consistent with the biological processes it mediates. Based on the analysis of celluar components and biological processes analysis, the *GmFtsH* genes may act as enzymes in chloroplast thylakoids to perform protein hydrolysis biological functions, which is consistent with the molecular functions of FtsH proteins in Arabidopsis [14], rice [15], and cyanobacterial [17]. The above results suggest that FtsH proteins mainly perform functions such as protein hydrolysis in chloroplasts, affecting leaf photosynthesis and thus plant growth status.

In addition, the *FtsH* gene family plays an important role in plant growth and development, and we identified some developmentally relevant cis-acting elements in the promoter of the *GmFtsH* genes, such as cell cycle regulatory elements, seed-specific regulatory elements, meristematic tissue expression elements, endosperm expression elements, and fenestrated cell differentiation elements. Previous research has demonstrated that *FtsH* genes are mostly expressed in a variety of organs, particularly leaves, floral organs, and seeds [13,24]. Transcriptomic data showed that most of the *GmFtsH* genes were highly expressed in leaves, and quantitative data showed that the expression of *GmFtsH* genes was significantly higher in new versus old leaves than in other tissues, which supported the hypothesis that *GmFtsH* genes play an important role in leaves as well as in chloroplast development. However, there were differences between the expression patterns of many *FtsH* genes in new and old leaves, which may be related to the biological roles of the genes. Meanwhile, we found that most of the *GmFtsH* genes were highly expressed in flowers, which can be hypothesized to be able to influence the flower development process.

Previous research showed that the FtsH complex connected to chloroplast development is made up of heterohexamers of (*FtsH1* and *FtsH5*) A-types and (*FtsH2* and *FtsH8*) B-types and that the simultaneous deletion of either A or B types led to lethal albinism in the plant, indicating that *FtsH* exerts its biological function in the form of a hexameric presence. It was found that homologs from different species usually have similar functions in biological processes. For example, *OsFtsH2*, a homolog of *AtFtsH2*, is involved in the development of rice chloroplasts. In the present study, we cloned *GmFtsH13*, a homolog of *AtFtsH5* and *AtFtsH1*, from “Williams 82”, and verified the subcellular localization of the homolog of *AtFtsH5* and *AtFtsH1*, as well as its function in the chloroplast.

In this study, we identified and characterized the *FtsH* gene family of *Glycine max* and evaluated its gene structure, conserved motifs, Cis-acting elements, phylogenetic relationships, chromosome location, collinearity, expression patterns, and subcellular localization. These results provide a basis and framework for further investigation of their potential in influencing the leaf and chloroplast development of *Glycine max*.

## 4. Materials and Methods

### 4.1. Identification of the GmFtsH Gene Family

The genomic sequences for *Oryza sativa*, *Glycine max*, *Arabidopsis thaliana*, and *Nicotiana tabacum* are available for download from the Ensembl Plants database (https://plants.ensembl.org/index.html, accessed on 30 April 2023). The Ensembl Plants database and the Phytozome database (https://phytozome-next.jgi.doe.gov/, accessed on 30 April 2023) were used to retrieve the protein sequences of the *FtsH* gene family from *Arabidopsis* [24], rice [34], and tobacco [13]. Using the FtsH protein sequence of *Arabidopsis thaliana* as a template, the candidate FtsH protein sequence of *Glycine max* (E-value ≤ 1 × 10^−5^) was identified using the blast program in TBtools-II (v2.012) [41]. The candidate genes were then identified by the CD-search function of NCBI (https://www.ncbi.nlm.nih.gov/Structure/bwrpsb/bwrpsb.cgi, accessed on 30 April 2023) and the Pfam database (http://pfam-legacy.xfam.org/, accessed on 30 April 2023) for conserved structural domains. The final protein candidates, designated GmFtsH1-25 and GmFtsHi1-GmFtsHi9 [42], were sequences with the FtsH protein-specific AAA structure domain (PF00004), peptidase M41 structural domain (PF01434), and FtsH Extracellular structural domain (PF06480) [42]. The protein sequences were then submitted to ExPasy (https://web. expasy. org/protparam/, accessed on 30 April 2023) [43] to predict their molecular weight, amino acid number, isoelectric point, and other information, using the online tool Softberry (http://linux1.softberry.com/berry.phtml, accessed on 30 April 2023) to predict subcellular localization [44].

### 4.2. Phylogenetic Relationship Analysis

Multiple sequence comparisons of *Glycine max*, *Arabidopsis thaliana*, *Oryza sativa*, and *Nicotiana tabacum* FtsH proteins were performed using Clustal W, and Neighbor-Jioning clustering analysis (Bootstrap = 1000) was performed using MEGAX64 (10.2.6) [45]. The phylogenetic tree was then embellished using the ChiPlot online tool (https://www.chiplot.online/#Phylogenetic-Tree, accessed on 1 May 2023).

### 4.3. Gene Structure and Conserved Motif Analysis

The exon–intron structure of *GmFtsHs* was determined using the online website Gene Structure Display Server (GSDS2.0, http://gsds.gao-lab.org/, accessed on 1 May 2023) [46]. And the conserved motifs of *GmFtsHs* were predicted by Multiple Expectation maximization for Motif Elicitation (MEME, https://meme-suite.org/meme/tools/meme, accessed on 1 May 2023), with the number of motifs set to 10, the minimum number of amino acids set to 6, and the maximum number of amino acids set to 50 [47]. Finally, TBtools-II (v2.012) was used to summarize and enhance the data.

### 4.4. Chromosome Location and Collinearity Analysis

To find out where *GmFtsHs* are located on the chromosomes, the soybean genome annotation files were loaded onto Gene Location Visualize from the GTF/GFF program of TBtools-II (v2.012). Gene duplication event analysis was completed using the Advanced Circos program and multi-species covariance analysis was completed using the Dual Synteny Plot program of TBtools-II (v2.012) [41].

### 4.5. Analysis of Promoter Cis-Acting Elements

The 2000 bp upstream sequences of *GmFtsHs* were obtained from the Phytozome database and analyzed for cis-acting elements using the PlantCARE database (http://bioinformatics.psb.ugent.be/webtools/plantcare/html/, accessed on 3 May 2023) [48]. The cis-acting elements were visualized by Microsoft Excel 2019.

### 4.6. GO Enrichment Analysis

GO functional category annotation of *GmFtsHs* was performed using the Functional Annotation Tool program of the DAVID database (https://david.ncifcrf.gov/, accessed on 6 May 2023) [49]. After performing the −log10 conversion on the *p* value values, the data were shown as bubble charts using the ChiPlot online tool (https://www.chiplot.online/bar plot_width category.html, accessed on 6 May 2023).

### 4.7. Expression Pattern Analysis

Based on the Phytozome database [50], the transcriptome information (FPKM values) for *GmFtsHs* in nine different tissues (leaves, rhizomes, root hairs, seed pods, stems, seeds, stem-tip meristematic tissues, flowers, and roots) of “*Glycine max* Wm82.a2.v1” was retrieved [50]. The expression levels of *GmFtsHs* were determined as the number of fragments per kilobase of transcript per million mapping reads. The number of transcriptional pieces per million mapped reads was used to define expression levels. Visual heat maps were completed based on the data using the software TBtools-II (v2.012).

### 4.8. Plant Materials and qRT-PCR Analysis

Soybean seeds of “Williams 82” provided by the Engineering Research Center of the Chinese Ministry of Education for Bioreactor and Pharmaceutical Development of Jilin Agricultural University were sown in the experimental field of Jilin Agricultural University. Flowers, stems, stem tips, new leaves, old leaves, seed pods (R5 stage), seed embryos (R5 stage), main roots, and lateral roots were collected from three well-grown plants after about 70 days. For a subsequent qRT-PCR analysis, samples were promptly frozen in liquid nitrogen or kept in a −80 °C refrigerator. RNAiso plus reagent (TAKARA, Beijing, China) was used to extract the samples’ total RNA, and agarose gel electrophoresis and the Nanodrop 2000 spectrophotometer (Implen, Germany) were used to determine the quantity and purity of the extracted RNA. And, using the MonScriptTM RTIII All-in-One Mix with dsDNase (Monad, MR0501M, Suzhou, China) reagent, the RNA was reverse transcribed into cDNA. In total, 14 *GmFtsH* genes were chosen for transcript pattern analysis. The *GmActin* gene was used as an internal reference to normalize gene expression and qRT-PCR primers were created using Primer5 (V5.5.0) (Appendix A). Utilizing the MonAmpTM ChemoHS qPCR Mix (Monad, MQ00401S, Suzhou, China) reagent, the qRT-PCR was carried out. In total, 3 biological replicates and 2 technical replicates were performed for each reaction, and the relative expression of genes was calculated by the 2^−∆∆Ct^ method.

### 4.9. Subcellular Localization Analysis

To determine the location of GmFtsH13 protein expression in cells, the full-length coding sequence without a terminator (CDS) was cloned to the pGDG vector, which contained a green fluorescent protein tag (mGFP) driven by a 35S promoter. For this purpose, homologous recombinant primers with *Xho* I and *Sal* I restriction sites were designed (Appendix A), the recombinant fragment products were recovered and linked to the pGDG vector, and the recombinant plasmids were transferred into Agrobacterium tumefaciens GV3101 using the freeze-thaw method. The pGDG-GmFtsH13 vector and the pGDG empty vector were infiltrated into the leaves of tobacco (*Nicotiana benthamiana*), respectively. Using an LSM 710 confocal laser scanning microscope (Zeiss, Jena, Germany), mGFP fluorescence images were seen 2–3 days afterward.

## 5. Conclusions

In this study, a comprehensive analysis of the *GmFtsH* gene family was performed. First, based on the evolutionary links of *FtsH* genes in Arabidopsis, rice, and tobacco, phylogenetic tree analysis was used to separate the 34 discovered *GmFtsH* genes into eight groups. Subsequently, exon–intron structure analysis, conserved domain analysis, and conserved motif analysis demonstrated that *GmFtsH* genes in the same group were highly conserved. Intraspecific and interspecific covariance analysis helped to study the evolutionary process of the *GmFtsH* gene family. In addition, promoter cis-acting element analysis, GO enrichment analysis, and gene expression analysis for different tissues of *Glycine max* indicated that the *GmFtsH* genes may play important roles in plant growth and development, especially for leaf and chloroplast development. Finally, we demonstrated that the GmFtsH13 protein was localized in chloroplasts. The above results indicate that the *GmFtsH* genes may play a role in the growth and development of soybeans, provide preliminary information for future studies of the *GmFtsH* gene family in soybean leaf development, and provide scientific references for further research on the biological functions and mechanisms of the *GmFtsH* genes.

## Figures and Tables

**Figure 1 ijms-24-16996-f001:**
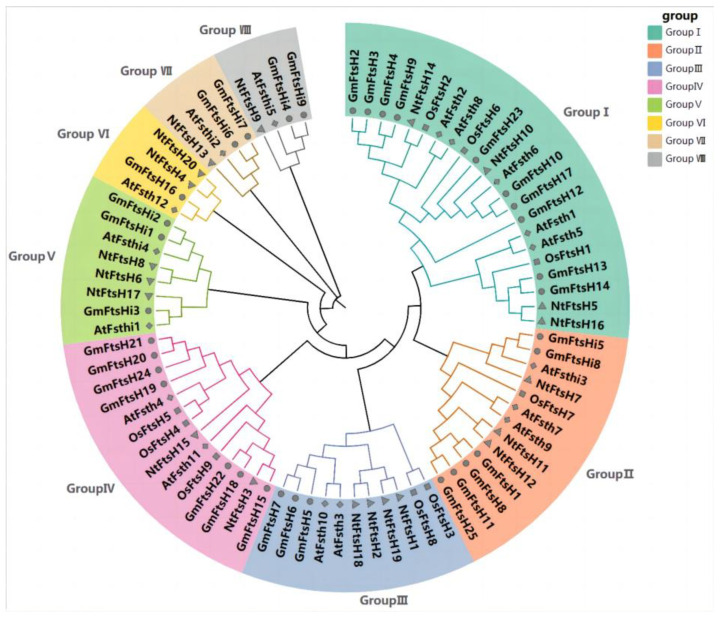
Phylogenetic tree of the *FtsH* gene family in *Glycine max*, *Oryza sativa*, *Arabidopsis thaliana*, and *Nicotiana tabacum*. A phylogenetic tree (1000 bootstrap replicates) was constructed using MEGAX64. Classes of different colors represent different groups. The genes beginning with “Gm” represent the genes of *Glycine max*, “At” represents the genes of *Arabidopsis thaliana*, “Os” represents the genes of *Oryza sativa*, and “Nt” represents the genes of *Nicotiana tabacum*.

**Figure 2 ijms-24-16996-f002:**
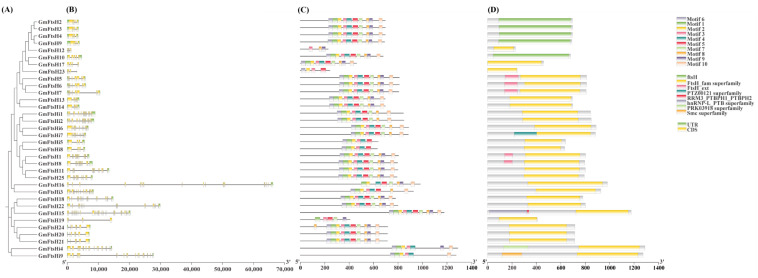
Analysis of the phylogenetic relationship, gene structure, conserved domains, and conserved motifs of *GmFtsH* genes. (**A**) Phylogenetic relationship analysis of *GmFtsH* genes. (**B**) Gene structure of *GmFtsH* genes—green boxes indicate 5′ or 3′ UTR regions, yellow boxes indicate exons, and black lines represent introns. (**C**) Conserved domains of *GmFtsH* genes. (**D**) Conserved motifs of *GmFtsH* genes.

**Figure 3 ijms-24-16996-f003:**
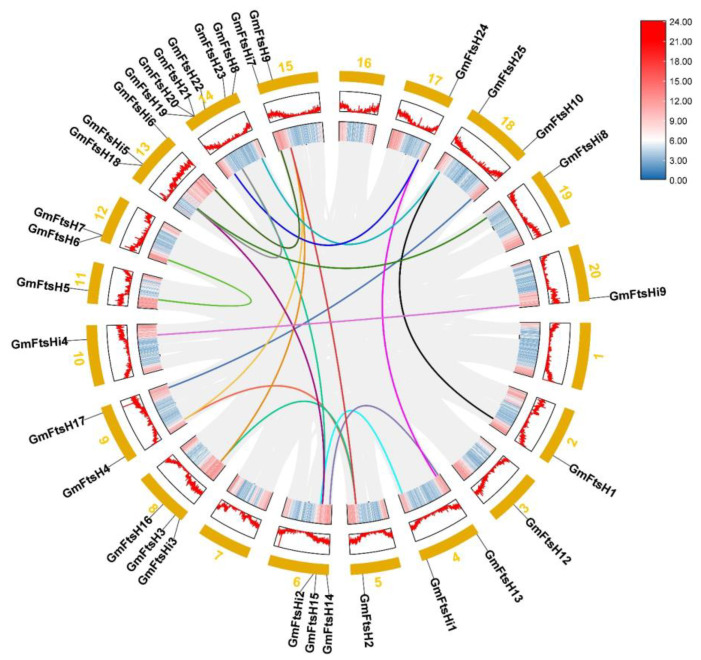
Chromosome location and gene duplication analysis of *GmFtsH* genes. Gray lines represent all collinear blocks in the genome of *Glycine max*; other colored lines represent duplicated *GmFtsH* gene pairs. The heat map in the inner circle indicates chromosomal gene density.

**Figure 4 ijms-24-16996-f004:**
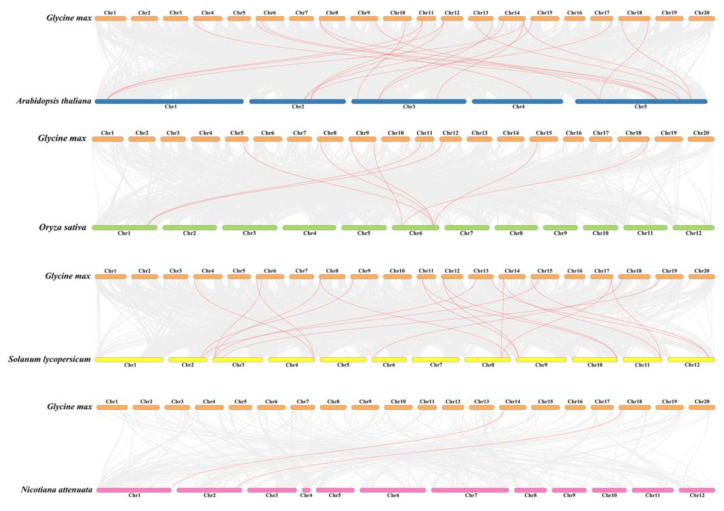
Collinearity analysis of the *FtsH* family of *Glycine max*, *Arabidopsis thaliana*, *Oryza sativa*, *Nicotiana attenuata*, and *Solanum lycopersicum*. Gray lines represent collinear blocks in *Glycine max* and other plant genomes and red lines highlight collinear *FtsH* gene pairs.

**Figure 5 ijms-24-16996-f005:**
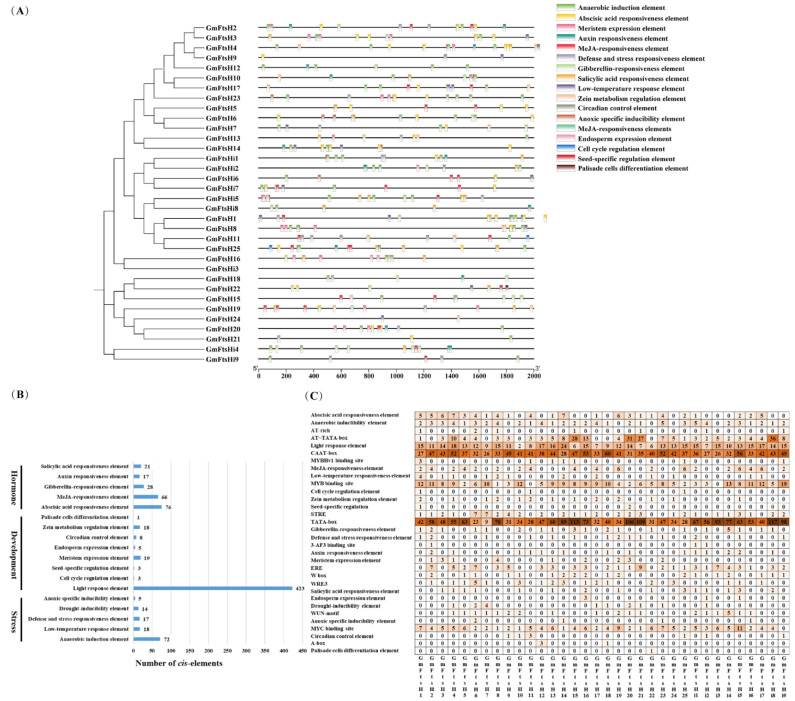
Schematic diagram of the cis-acting element analysis of the *GmFtsH* genes promoter. (**A**) The black line indicates the length of the *GmFtsH* genes promoter. Rectangular boxes with different colored boxes represent different types of cis-acting elements. The clustering of *GmFtsH* genes was based on the phylogenetic tree shown in Figure 2. (**B**) The bar chart shows the different types and numbers of three cis-acting elements contained in the *GmFtsH* genes promoter. (**C**) Shades of orange and numbers in the grid represent the number of corresponding cis-acting elements, and the darker the orange color, the more cis-acting elements there are.

**Figure 6 ijms-24-16996-f006:**
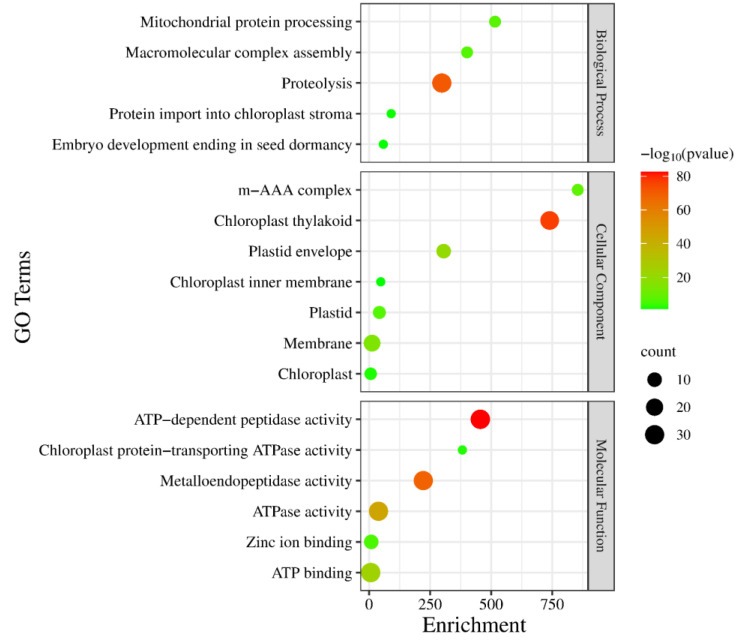
GO enrichment analysis of *GmFtsH* genes. Gene ontology (GO) enrichment analyses were divided into three main categories: molecular function (MF), cellular component (CC), and biological process (BP). GO terms associated with a *p* value < 0.05 were identified as significant.

**Figure 7 ijms-24-16996-f007:**
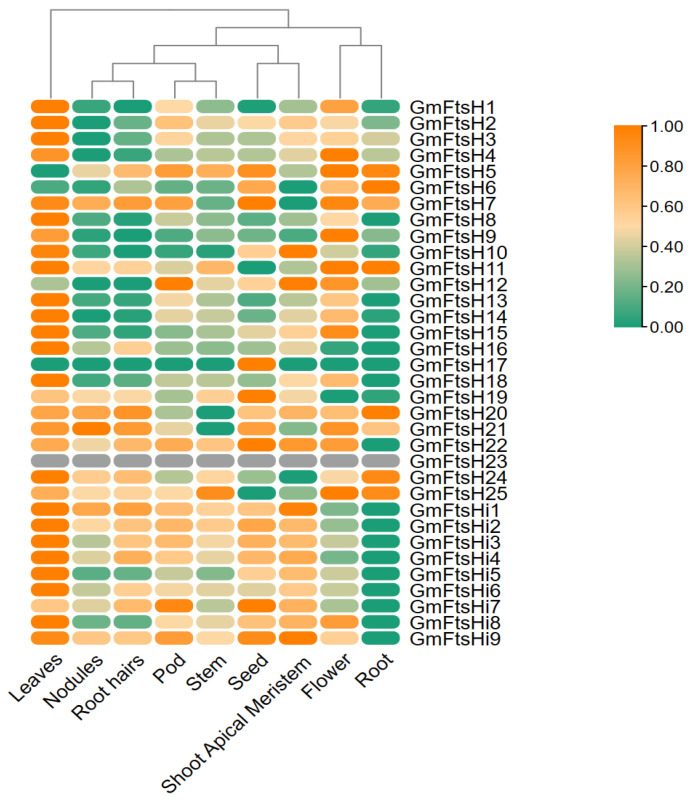
Expression heat map of *GmFtsH* genes in different tissues of *Glycine max*. The relative expression level is represented on the heat map’s right side by a color scale, and a gradient from dark green to orange-yellow shows an increase in expression level.

**Figure 8 ijms-24-16996-f008:**
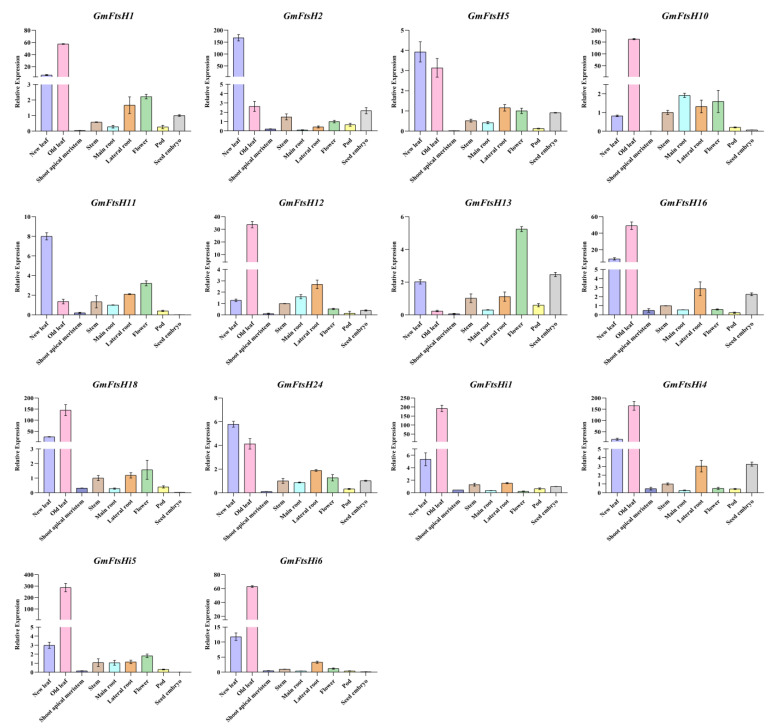
Expression profiles of different tissues of *GmFtsH* genes by qRT-PCR. The expression profiles of 14 *GmFtsH* genes at different developmental stages were determined by qRT-PCR analysis. The expression analysis was normalized using the *GmActin* gene as an internal reference. Error bars were estimated based on the differences in expression patterns of three independent replicates.

**Figure 9 ijms-24-16996-f009:**
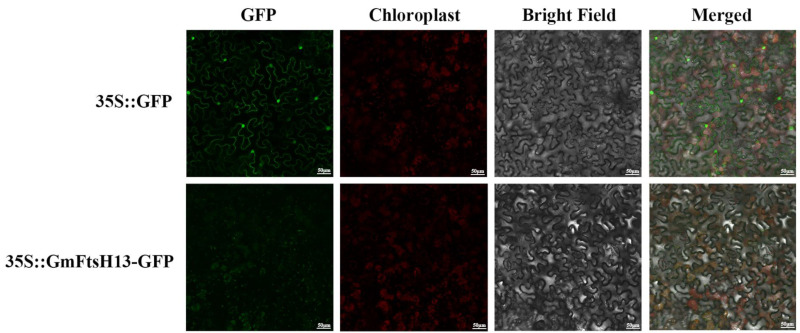
Subcellular localization of GmFtsH13. The control (35S::GFP) and fusion (35S::GmFtsH13-GFP) vectors were separately expressed in tobacco (*Nicotiana benthamiana*) leaves. GFP indicates green fluorescent protein signal. Merged indicates the merged signal. Scale bar = 50 µm.

**Table 1 ijms-24-16996-t001:** Analysis of physicochemical properties of *GmFtsH* genes.

Gene	Gene_Locus	Number of Amino Acids	Molecular Weight (Average)	PI	Instability Index	Aliphatic Index	Grand Average of Hydropathicity	Subcellular Localization
*GmFtsH1*	Glyma.02G225300	803	86,930.02	8.55	37.17	92.76	−0.193	Extr
*GmFtsH2*	Glyma.05G132000	695	74,816.31	5.56	33.88	91.51	−0.139	Mito
*GmFtsH3*	Glyma.08G086600	696	74,980.52	5.73	33.92	91.11	−0.144	Mito
*GmFtsH4*	Glyma.09G052600	695	74,631.55	5.89	34.52	94.4	−0.058	Mito
*GmFtsH5*	Glyma.11G137700	810	89,324.67	8.51	29.58	81.4	−0.37	Mito
*GmFtsH6*	Glyma.12G061200	810	89,316.76	7.96	32.43	83.65	−0.331	Mito
*GmFtsH7*	Glyma.12G061400	806	88,528.65	6.72	31.3	83.14	−0.305	Mito
*GmFtsH8*	Glyma.14G192100	795	86,419.51	8.8	38	91.25	−0.224	Extr
*GmFtsH9*	Glyma.15G158900	690	74,096.77	5.89	35.64	91.29	−0.106	Mito
*GmFtsH10*	Glyma.18G259700	678	73,349.2	5.88	37.71	95.53	−0.126	Mito
*GmFtsH11*	Glyma.U026800	799	86,937.02	8.38	41.56	91.05	−0.195	Extr
*GmFtsH12*	Glyma.03G089200	228	25,504.24	5.99	30.46	90.66	−0.314	Chlo
*GmFtsH13*	Glyma.04G019100	694	74,160.56	5.87	37.68	91.96	−0.092	Mito
*GmFtsH14*	Glyma.06G019400	696	74,393.95	5.93	37.01	91.82	−0.077	Mito
*GmFtsH15*	Glyma.06G126000	1176	128,842.86	5.89	49.12	91.74	−0.206	Extr
*GmFtsH16*	Glyma.08G237500	982	113,514.78	6.8	47.15	87.91	−0.397	Extr
*GmFtsH17*	Glyma.09G237900	459	49,413.52	5.24	31.04	95.82	−0.132	Mito
*GmFtsH18*	Glyma.13G041700	779	85,157.15	5.3	50.51	90.69	−0.248	Mito
*GmFtsH19*	Glyma.14G095700	406	43,905.2	5.03	37.47	90.05	−0.105	Mito
*GmFtsH20*	Glyma.14G096000	713	76,969.75	8.21	42.64	87.91	−0.237	Mito
*GmFtsH21*	Glyma.14G096100	713	77,151.98	8.89	43.02	88.04	−0.249	Mito
*GmFtsH22*	Glyma.14G124900	799	87,074.63	5.57	47.92	91.21	−0.182	Mito
*GmFtsH23*	Glyma.14G162200	241	26,544.18	5.01	32.74	92.57	−0.3	Mito
*GmFtsH24*	Glyma.17G228100	714	77,332.22	8.8	46.58	88.47	−0.243	Mito
*GmFtsH25*	Glyma.18G065600	792	86,240.29	8.38	44.66	91.96	−0.194	Extr
*GmFtsHi1*	Glyma.04G213800	843	94,795.31	5.65	44.6	89.6	−0.278	Extr
*GmFtsHi2*	Glyma.06G152500	847	95,554.22	5.86	43.7	87.91	−0.329	Extr
*GmFtsHi3*	Glyma.08G023900	926	104,155.24	9.04	41.35	96.11	−0.25	Extr
*GmFtsHi4*	Glyma.10G164800	1288	147,844.1	7.58	43.66	88.73	−0.37	Extr
*GmFtsHi5*	Glyma.13G049800	638	70,631.46	9.46	45.98	90.3	−0.188	Extr
*GmFtsHi6*	Glyma.13G355400	887	100,936.97	9.42	43.53	83.25	−0.545	Extr
*GmFtsHi7*	Glyma.15G018800	883	100,161.31	9.43	44.26	83.17	−0.516	Extr
*GmFtsHi8*	Glyma.19G040200	631	69,692.62	9.55	43.63	90.7	−0.174	Extr
*GmFtsHi9*	Glyma.20G227000	1274	146,660.97	8.33	41.76	89.84	−0.357	Extr

“Extr” represents extracellular, “Mito” represents mitochondrion, “Chlo” represents chloroplast.

## Data Availability

Data are contained within the article and Appendix A.

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
