# Peer review of "Genome-Wide Identification and Comprehensive Analysis of the *FtsH* Gene Family in Soybean (*Glycine max*)"

_ijms, 2023, doi:10.3390/ijms242316996_

Round 1
Reviewer 1 Report
Comments and Suggestions for Authors
The results of the study provide a basis or foundation for determining the function of GmFtsH genes. Additionally, it indicates that these findings contribute to a better understanding of the regulatory network involved in the development of soybean leaves. While the manuscript is well written, there are a few minor issues that require the authors’ attention:
In lines 115-117, the authors mentioned that the “FtsH genes from the four species were present 115 in Group I, II, III, and IV, implying that these members of the four plant species may have 116 undergone similar evolutionary patterns”. However, they have not stated the rationale behind the grouping. I would recommend the authors to add the rationale.
Fig. 2 is very broad and has a lot of information in it. I would recommend increasing the font size of the figure so the readers can understand what each color box represents.
Fig. 6 just shows the 3D structural modeling of different proteins. In my opinion, those structures are not required if the homology modeling has not been solved experimentally (X-ray, NMR).
Fig. 9 shows cut in between in several graphs. What does that imply?
Reviewer 2 Report
Comments and Suggestions for Authors
The manuscript titled "Genome-wide identification and comprehensive analysis of the FtsH gene family in soybean (Glycine max)" is a good bioinformatics work. It is mainly based on the results contained in online databases. My only reservations are the lack of a clearly formulated research goal at the end of the Introduction and a conclusion referring to the goal at the end of the manuscript. The manuscript meets the requirements of a scientific article and I recommend it for minor revision.
